# Using Glittr.org to find, compare and re-use online materials for training and education

**Geert van Geest**[1,2]*, **Yann Haefliger**[1], **Monique Zahn-Zabal**[1], **Patricia M. Palagi**[1]

**1** Swiss Institute of Bioinformatics, Quartier Sorge—Bâtiment Amphipôle, Lausanne, Switzerland,
**2** Interfaculty Bioinformatics Unit, University of Bern, Bern, Switzerland

* geert.vangeest@sib.swiss

**Data Availability Statement:** All data analyzed in the manuscript are accessible through https://glittr.org and can be programmatically accessed through the REST API at https://glittr.org/api/repositories.

## Abstract

A wealth of excellent training and educational materials for the computational life sciences are scattered around the Internet, but they can be hard to find. Many materials reside in public Git repositories that are hosted on platforms such as GitHub and GitLab. Glittr.org is a manually curated database of Git repositories, which enables users to find educational materials that would otherwise be hard to identify. With the application, users can search and compare educational materials based on topic and author, but also on engagement metrics such as stargazers (bookmarks) and recency (days since last commit). Glittr.org currently contains 664 entries, which are assigned to six different categories within the domain of computational life sciences. By analysing the database, we reveal insights in the availability of materials per topic, collaboration patterns of developers, and licensing practices. This knowledge helps to understand in which areas open educational materials are scant, the importance of Git for collaboration on educational materials and how licensing can be improved to enhance sharing and reuse. Taken together, we show that Glittr.org contains a wealth of connected and openly available metadata. Therefore, it enhances adherence to the FAIR (Findable, Accessible, Interoperable, Reusable) principles, which benefits learners, teachers and trainers in the entire life sciences community and beyond.

## Introduction

The life sciences domain is constantly growing and evolving, requiring professionals to be trained to stay up to date with the most relevant skills and knowledge. To facilitate this, various organizations deliver courses on a large range of topics in many different formats. Many course providers dedicate themselves to actively sharing their materials with the public, resulting in a growing wealth of public training and educational materials available across the web. This greatly enhances the possibility to re-use these educational materials, which is invaluable for both learners and lesson developers; learners can use the material on their own time to obtain required skills, while lesson developers can use the existing material to build their courses further upon existing ones.

To help developers make their shared material easier to find and re-use, the FAIR principles [1–5] provide guidelines on organizing their materials and metadata. The way educational material metadata are represented depends on the platform where they are hosted. Currently,

**Funding:** The author(s) received no specific funding for this work.

**Competing interests:** The authors have declared that no competing interest exist.

educational materials are hosted in diverse locations, like organizational websites, cloud storage, learning management systems, and Git platforms (e.g., GitHub, GitLab). Therefore, metadata representation often does not conform to standards, making it difficult to parse and utilize it effectively. Aggregation and indexation require such metadata to be standardized and help representing previously unstructured information in a structured, machine-readable way. They therefore enhance findability, re-usability, and integration of metadata over multiple platforms.

Registries that currently aggregate and index educational materials in the life sciences include the Open Educational Resources (OER) Commons [6], the training portal of the Global Organisation for Bioinformatics Learning, Education and Training (GOBLET) [7], and ELIXIR's online training registry Training eSupport System (TeSS) [8]. These registries can accommodate any digital object related to education that are accessible through an URL, making them potentially comprehensive. However, this flexibility also comes with disadvantages. As not every webpage contains standardized metadata, it requires investments from the educational material developer to manually enter metadata in the registry or to automate metadata provision, for example by implementing a standardized metadata markup like Bioschemas [9, 10].

Using public Git repositories for developing and hosting educational materials is gaining popularity, especially for materials related to (bio)informatics and data science. It is generally recommended for reproducibility [11] and adopted by many institutions, communities, and individuals (e.g. the Carpentries [12], Galaxy training network [13], the National Bioinformatics Infrastructure Sweden (NBIS), and the SIB Swiss Institute of Bioinformatics). Having educational materials in public Git repositories has major advantages. First, using Git encourages community-driven development. Git is version-control software, and as such, can be used to track changes in files and associate these changes to individuals with distinct levels of permissions. This makes it ideal for co-development and re-use of materials. Using Git drives community engagement by supporting direct feedback, suggestions, and corrections. Secondly, the infrastructure around the most popular platform for public Git repositories, GitHub (http://github.com) and GitLab (http://about.gitlab.com), are standardized and commonly used. Many people have the know-how to use version control, so the threshold to use or learn how to use Git for educational material is low–especially in communities related to the informatics domain. Thirdly, public Git repositories offer standardized retrieval of a wealth of metadata. Authorship, activity logs, and popularity metrics are all recorded and made accessible through a single point of entry.

Although both GitHub and GitLab have a sophisticated search functionality, it is challenging to identify and compare repositories with educational materials on a topic of interest. The findability of repositories with educational materials depends on the effort of the author to describe the repository, for example in the repository description, the name, the associated website, and repository tags. Since not all authors annotate their repositories extensively and the way developers annotate repositories is not standardized, many repositories with excellent educational materials remain hard to find.

This article introduces Glittr.org, a novel web-based resource that enhances the discovery and comparison of reusable educational materials on topics related to the computational life sciences that are hosted as public Git repositories. This database of Git repositories is publicly maintained, and the listed repositories are made findable and comparable by tagging them with standardized topics and reporting their popularity and recency. This helps both learners and educators with finding valuable educational materials that fit their needs. Glittr.org supports the adherence to the FAIR principles by standardising metadata and making it accessible through REST API endpoints, further fostering findability and reusability of educational materials. Lastly, because many of the topics of educational materials represented in Glittr.org

encompass other domains, Glittr.org is also useful to anyone working in informatics, data science, or machine learning.

## Materials and methods

### Eligibility and collection of educational materials

Glittr.org is a publicly contributed and maintained list of educational materials for the computational life sciences from anywhere in the world and in any language. The database consists of public GitHub or GitLab repositories. The teaching aim of the material should be a general topic in the computational life sciences, not tool usage–meaning that inclusion of tutorials on using single software are discouraged. The reason for this is to ensure Glittr.org provides an overview of metadata on educational materials, and not on metadata related to software. The quality of the educational materials is not evaluated. Repositories that are potentially eligible to be part of Glittr.org are identified from existing collections, internet searches, specific searches on GitHub, (social media) promotions, and user contributions through the web form. Repository metadata are retrieved from GitHub and GitLab. Among those are the repository name, URL, repository description, author and contributor names, their profile and website, the number of stargazers (bookmarks), the date of the last push, and the license. These are all programmatically accessible from the REST APIs of GitHub and GitLab.

### Usage of existing ontologies

To enable categorization and improve search functionality in Glittr.org, the educational content in the repositories is described with tags, for instance 'RNA-seq', 'Python', etc. These tags are based on concepts found in existing ontologies whenever possible. Using ontologies greatly enhances standardization and therefore knowledge sharing. However, educational material for the computational life sciences can cover a very wide range of topics, and using a single ontology to describe them can be problematic. Glittr.org therefore uses multiple ontologies, aiming to give a topic description which is as rich as possible for each repository. Currently, the following ontologies are used: EDAM [14], Data Science Education Ontology (http://www.pagestudy.org/DSEO/), Experimental Factor Ontology [15], and FAIR Data Train Ontology (https://bioportal.bioontology.org/ontologies/FDT-O). If a term is considered relevant, but not in any of these ontologies, a term will be used from any other ontology on BioPortal [16], a software on bio.tools [17] or an item at Wikidata [18]. By using multiple ontologies, we have the flexibility to annotate the wide range of educational material, but still have good level of standardization in relation to the relevant domain. If a tag is required but not yet available in Glittr.org, contributors can request it through the contribution form, and a curator from the Glittr.org team will identify it from an ontology and add the tag if appropriate.

### Usage of tags and categories

Each repository is associated with one or more tags as decided by the contributor, and these tags are checked by a curator from the Glittr.org team. Here, the order of tags matters, meaning that the tag that describes the content best is first (the 'main tag') and is then followed by other relevant tags all with equal importance. For example, for a single cell transcriptomics course in R, the tag order could be 'Single-cell sequencing', 'R', 'transcriptomics', and 'RNA-seq'. The first and therefore main tag would be 'Single-cell sequencing', which is used to categorize the material, while the others have equal importance and are used to further describe the material. To improve searchability, each tag falls within a broader category, which are curated and maintained by the Glittr.org team. Currently, these are: (1) Scripting and

languages, (2) Computational methods and pipelines, (3) Omics analysis, (4) Reproducibility and data management, (5) Statistics and machine learning, and (6) Others. The main tag of the above-mentioned single cell course is part of the category 'Omics analysis' and so the course material falls within this category.

## Web application

The Glittr.org web application is created using the Laravel (http://laravel.com/) framework and is written in PHP. Repository metadata is actively updated twice daily using Laravel jobs. This approach enables handling of potential issues by implementing delayed retries if the GitHub/GitLab APIs are temporarily unavailable. Each metadata update for a single repository constitutes a dedicated Laravel job, with the overall management of all repository metadata updates performed with the Laravel Haystack package (http://docs.laravel-haystack.dev/). This package ensures job validation and sends email notifications in case of errors.

All the reactivity supporting list functionalities such as filtering, pagination, and sorting is based on the Livewire (http://livewire.laravel.com/) framework. The JSON API endpoints for repositories and Bioschemas are generated with the Laravel-query-builder (http://spatie.be/docs/laravel-query-builder/v5/introduction) package developed by Spatie (http://spatie.be/open-source). This ensures the query parameters and output results follow the JSON API specification as closely as possible. The backend to manage data (topics and categories, repositories, submissions) is created with the package Jetstream (http://jetstream.laravel.com/introduction.html).

All source code is accessible at http://github.com/sib-swiss/glittr and the website is available through https://glittr.org. The application is a SIB Training resource and is hosted on servers at SIB. As an institute with a long history of working with database life cycles, we are committed to sustaining Glittr.org for as long it is serving the community by aligning to the TRUST principles [19].

## User interface

A snapshot of the front-end website is found in Fig 1. The user can easily access both the repository with course materials and the course website (when applicable). In addition, a link to public information about the author or organization that created the material is provided to retrieve more author information. The topics (tags) covered by the material allows users to quickly find course material relevant to their search. The number of stargazers estimates the popularity of the repository, while the number of days since the last push provides an indication of the recency of the material in the repository. The license which applies to the repository contents indicates to educators wishing to re-use the material what conditions apply.

## REST API

The application currently has three REST API endpoints in which data stored by Glittr.org can be programmatically accessed: api/repositories, api/tags, and api/bioschemas. The api/repositories endpoint contains all repositories including their metadata as presented on the website. The repositories/tags endpoint contains tag metadata organized by their corresponding category. Tag metadata consists of the ontology, ontology class, and description. The api/bioschemas endpoint contains repository and tag metadata according to the Bioschemas [9] TrainingMaterial profile [10] (http://bioschemas.org/profiles/TrainingMaterial).

## Statistics on the database

To give an overview and statistics of all the metadata stored on Glittr.org, we used the R httr2 package to access the Glittr.org REST API (see above). Public information on authors, and

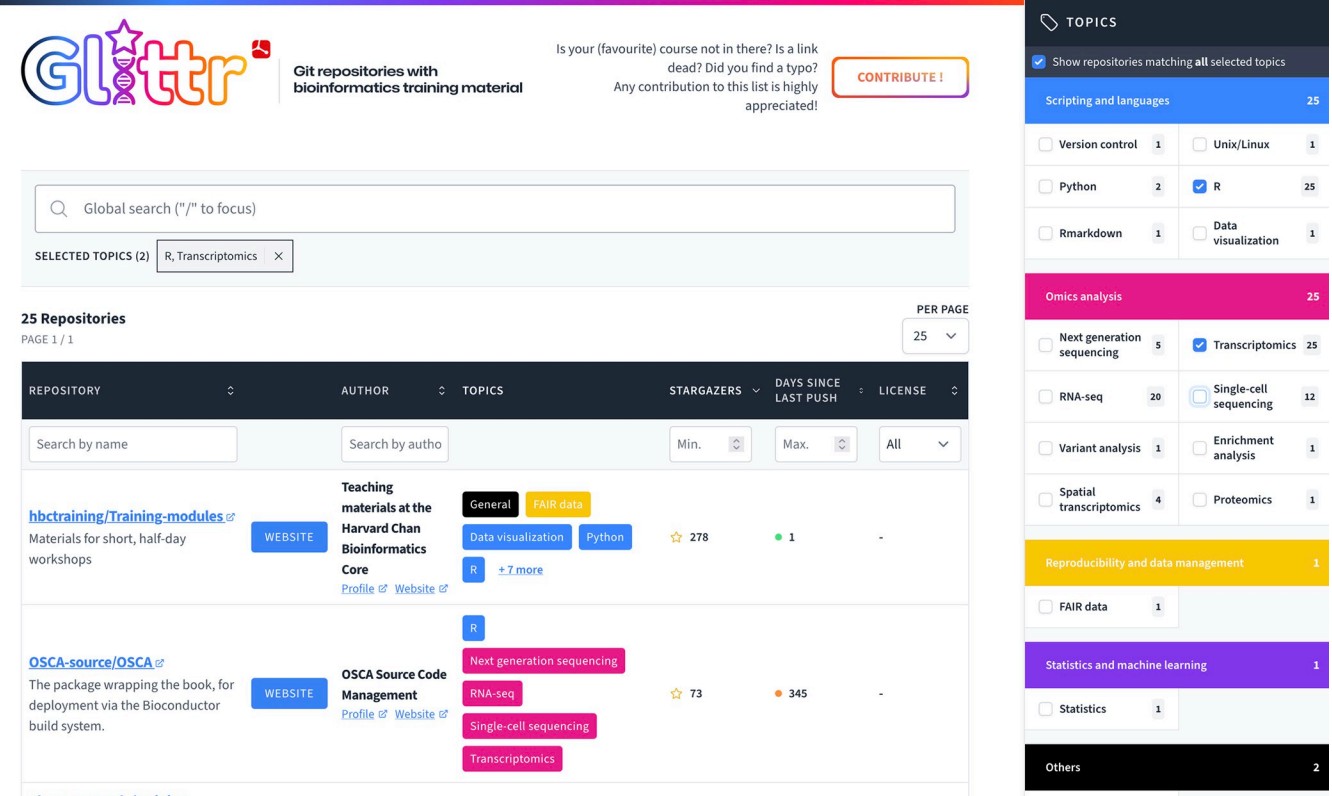

**Fig 1. Snapshot of Glittr.org.** Two tags are selected, 'Transcriptomics' and 'R', showing the top 2 of 25 repositories with both tags. By default, results are sorted based on number of 'stargazers', i.e. bookmarks.

number of contributors was extracted from the GitHub REST API. Location of authors and organization was extracted from the 'location' field within the users API. This string was translated in a country with the R package geocode, which has wrappers for the Google Geocoding API. The source code for acquiring the statistics can be found at http://github.com/sib-swiss/glittr-stats and the report can be found at http://sib-swiss.github.io/glittr-stats/ (updated weekly).

## Results and discussion

There are currently 664 repositories listed on Glittr.org. By combining data from our REST APIs and those of GitHub and GitLab we can give a detailed insight in these repositories. In the first part of this section, we show the kind of repositories with educational material that are currently available, for which topics educational materials exist, what licenses are used, and by whom the repositories are created. In the second part we present several use-cases on how different types of users can interact with Glittr.org to show how educators, learners and developers can use different functionalities of Glittr.org.

### Training material topics

Repositories on Glittr.org are categorized in six categories (as mentioned above) and tagged with 58 different topics. All are connected to computational life sciences, but many are not specific to it: for example, programming languages are relevant to informatics in general, and statistics to all domains related to science. Therefore, using Glittr.org is relevant to a broader

public than just persons working in the domain of the computational life sciences. In this section we give a brief overview of popular categories and topics of educational materials on GitHub and GitLab that have been added to Glittr.org.

By far, most repositories currently on Glittr.org have 'Scripting and languages' as main category– 315 in total (Fig 2A). Within this category, R [20] is the most frequent tag with 268 repositories, followed by Python with 100 repositories (Fig 2C). This suggests that the R language is a popular language to teach and therefore probably to learn. However, the popularity of R might not be the only reason; the ease with which one can host static websites within the R framework, with e.g. bookdown [21] or Quarto (http://quarto.org/), could also explain this.

The category 'Omics analysis' is specific to the domain of computational life sciences, and many repositories fall within this category. 'Transcriptomics' is the most frequent tag (86 repositories), followed by 'RNA-seq', suggesting an important place for transcriptomics-related courses in current bioinformatics curricula. Transcriptomics analysis skills are popular to acquire, likely because it is one of the most used applications of next generation sequencing, it is applied in many domains in the life sciences [22] and each experiment requires its own analysis choices. Its demand is well served by the vast amount of material available on transcriptomics.

There are 34 repositories on 'Data management', representing its growing importance within the scientific community regarding trends in open and FAIR data. Using public Git repositories to develop educational materials on data management related topics is likely to be of particular interest because using Git supports transparency and community-based development. Efforts that support open and reproducible science like the ELIXIR Research Data Management (RDM) community [23], European Open Science Cloud (EOSC) [24] and other research infrastructures likely influence this trend towards using Git repositories for educational materials both on data management and other topics.

The number of repositories and their diversity show that there is a wealth of public repositories with educational material on computational life sciences. For many topics, there are high-quality materials in different formats, giving lesson developers and learners many possibilities to re-use existing educational materials. Glittr.org organizes this wealth of materials, making them better findable and therefore available for re-use.

## Collaboration

An important advantage of using version control for educational materials is that it supports collaboration and the reuse of existing material. This is also represented in the number of collaborators per repository that are on Glittr.org. Most repositories (80.1%) have more than one contributor, meaning the repository is developed by more people than only the original author. Most repositories have only a few contributors, between 1 and 5 (60.3%). However, quite a large fraction (24.8%) has more than 10 contributors, showing that these repositories gain from a strong community involvement and therefore increased (re-)usage and quality.

Repositories in the category 'Reproducibility and data management' and 'Computational methods and pipelines' have the largest fraction of repositories developed by a large community (> 50 contributors; Fig 2B). Repositories with a large contributor base within these categories are e.g. elixir-europe/rdmkit—which is a community-created website with research data management (RDM) best practices [25], galaxyproject/training-material–the Galaxy training network repository with all educational materials related to the workflow framework Galaxy [13]—or are forked from a repository with a large community base (e.g. a template from the carpentries [12]). This shows that large and important initiatives rely on Git for the development of educational materials and that these thrive by cooperation.

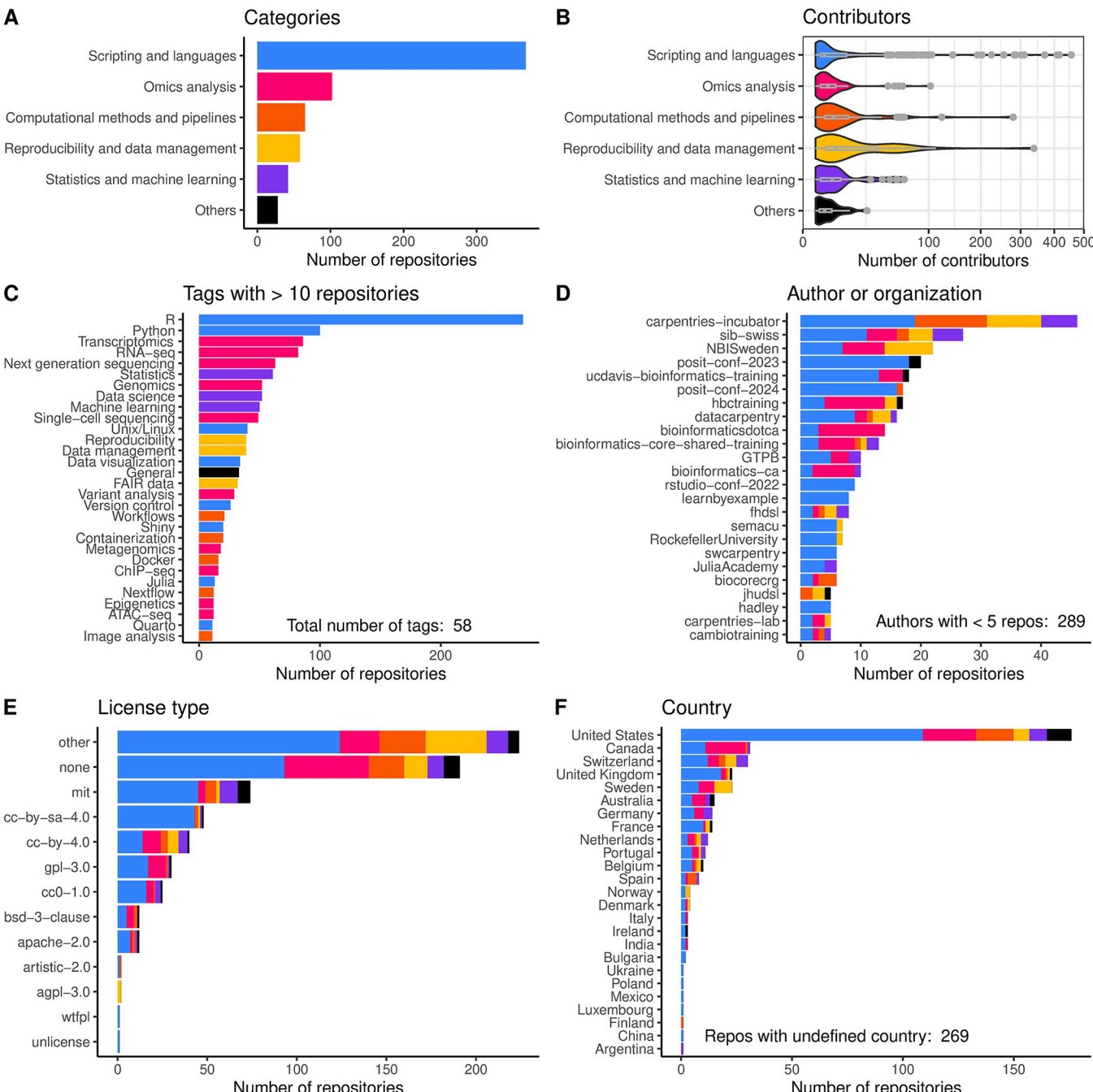

**Fig 2. Key Glittr.org statistics displayed by category.** Overview of number of repositories by category (A), distribution of number of contributors per repository by category (square root transformed; B), number of repositories per tag (C), per author or organization (D), per license (E) and per country (F). Colours are according to category as depicted in (A). In C) only the tags with more than 10 repositories are displayed and in D) only the authors/organizations with five or more repositories. At time of writing Glittr.org contained repositories authored by 242 authors/organizations with less than five repositories. For A), D), E) and F) the plots are coloured according to the category of the main (i.e. first) tag.

## Authors and organizations

The origin of repositories on Glittr.org is very diverse; currently, the 664 repositories originate from 313 different GitHub/GitLab authors or organizations. The organization with most

repositories is the carpentries-incubator [12], followed by two national institutes, NBIS and SIB (Fig 2C). Obviously, the distribution of repository category varies per organization/author. Bioinformatics institutes, like bioinformatics.ca, Instituto Gulbenkian de Ciência (GTPB), CRUK CI (bioinformatics-core-shared-training), Harvard bioinformatics core (hbctraining), UC Davis Bioinformatics Core (ucdavis-bioinformatics-training), SIB and NBIS, are more focused on computational biology and have more courses in the category 'Omics analysis'. Other namespaces with many repositories like carpentries incubator [12], the '23 and '24 posit conferences (posit-conf-2023 and posit-conf-2024) and learn by example (learnbyexample) are focused on computer skills and therefore have mostly repositories on 'Scripting and languages' and 'Computational methods and pipelines' (Fig 2C).

In addition to the number of authors, the number of countries listed in the authors' affiliations is also diverse. Current repositories on Glittr.org originate from 26 different countries. Most repositories with the main author or organization having a country defined in its profile originate from the United States, followed by Switzerland and Canada (Fig 2F). This shows that most educational materials originate from the Western and particularly English-speaking parts of the world, and possibly from regions that incentivize Open Science and education. English is currently the 'lingua franca' of the academic world [26], and most educational materials on the computational life sciences reflect that. Connections between the academic ecosystem and educational materials not written in English can be literally lost in translation, and therefore hard to identify. However, there are many communities that demand non-English educational material on computational life sciences, and we are open to serving these communities by listing them on Glittr.org. Organization of the curation of repositories in other languages is more challenging because it relies on reading and understanding the content, but we encourage initiatives in this direction.

## Licensing

Although licensing is an important way to enable and encourage re-use, licensing of public educational materials on GitHub and GitLab is not very standardized. Fig 2B gives an overview of licenses per repository that could be recognized as such. For many repositories (191; 28.9%), there is no license detected, and for 224 (33.8%) repositories the license could not be automatically recognized. The most popular recognized license is the MIT license, a software license. The copyright licenses based on Creative Commons (CC) are also quite popular, 113 repositories have a license recognized as such. Since educational materials generally consists of text, presentations, and exercises, and not software, it is usually more appropriate to use a copyright license instead of a software license [4, 11]. It therefore seems that many developers of educational materials might want to (re-)consider their license.

## Use cases

Glittr.org has been developed for use by computational life sciences learners, educators, lesson developers and database developers. To showcase different applications of the resource, we present four different use cases that we have envisaged for Glittr.org. The first three use cases illustrate the ease of use, as only a single click is necessary to obtain an answer, while the last one shows how to retrieve information programmatically.

**A student looking for educational materials.** Our first use case is that of a student looking for educational materials, for instance someone wishing to learn the programming language Julia. This high-level, general-purpose dynamic programming language used for numerical analysis and computational science is usually offered in university and higher order education curricula. Using the 'Filter by topic' option to select Julia under the 'Scripting and Languages'

section results in repositories containing training materials on the topic. The top result is the most popular as, by default, the results are displayed in order of decreasing number of stargazers. It is also possible to sort by increasing number of days since the last push to find material which has been recently updated. The link to the website takes the student directly to the page showing the different options to get started with Julia. Glittr.org thus allows students to quickly access the most popular, public course material on the subject.

**An educator looking for material for re-use.**   This use case is that of a trainer who will be teaching a course on Snakemake and is looking for materials that can be re-used. Snakemake is a workflow management system to create reproducible and scalable data analyses. Selecting Snakemake in the Computational methods and pipelines section of the 'Filter by topic' option retrieves several repositories with materials to teach Snakemake. By looking at the license column, the trainer can easily spot that there are two repositories that contain material that have a re-use permissive license, one under the Creative Commons Attribution-ShareAlike 4.0 International license (CC BY-SA 4.0) and another under the GNU General Public License version 3 license (pl-3.0). For other repositories in the list, the license could not be automatically identified, and therefore the license has been indicated as 'other'. For these repositories some further investigation is needed before they can be re-used. The trainer has quickly found re-usable material on the topic of interest using the Glittr.org website.

**An educator sharing their own materials.**   The third use case is that of a trainer who has developed a course on a hot topic in the computational life sciences on GitHub or GitLab and wishes to share the material. Contributions to the Glittr.org website from the training community are welcome and encouraged. The Contribute button on the top right of the home page (Fig 1) links to a simple form. By completing the submission form, the trainer's repository will be added (or updated) by a member of the Glittr.org team to the collection provided it meets the inclusion criteria (public GitHub or GitLab repositories containing educational material on the computational life sciences). The trainer's material is now findable by being included in the Glittr.org website. Obviously, this does not prevent the trainer from posting the material in other registries which is, in fact, recommended by the FAIR principles (the more the material is registered, more chances are that they are findable).

**A database developer wanting to programmatically access Glittr.org.**   A final use case is that of a developer who wishes to programmatically access one or more subsets of educational materials repositories on Glittr.org. This can easily be done using the API endpoints (http://github.com/sib-swiss/glittr#api) and applying suitable filters. For example, the data for the graphs in Fig 2 are all extracted from the Glittr.org API with the R packages httr2 [27]. The Bioschemas API endpoint allows portals that can ingest Bioschemas markup to re-use Glittr.org data. For example, ELIXIR TeSS [8], currently ingests this metadata (http://tess.elixir-europe.org/content_providers/glittr-org).

## Conclusion

As bioinformatics training providers, we saw the need for a simple-to-use, web-based listing of git repositories with educational materials on computational life sciences. Since its establishment 1.5 years ago, more than 650 repositories with educational materials on computational life sciences with a diverse set of topics were included in Glittr.org. By focussing only on Git repositories, Glittr.org can automatically display relevant information like popularity, license and recency that would otherwise not be possible. As illustrated by the five use cases, the resource helps with educational material selection by students and educators and supports the integration of educational material databases. All the information is available programmatically via the API and Bioschemas markup thereby enabling other training registries to freely

update their existing entries or add entries not yet present in their database. In the future, we hope to develop more strategies for Glittr.org that encourage interoperability by establishing connections with other life sciences databases. This will further promote the findability, accessibility, interoperability, and re-usability (FAIRness) of educational materials in the life sciences and other informatics-driven fields.

## Author Contributions

**Conceptualization:** Geert van Geest, Monique Zahn-Zabal, Patricia M. Palagi.

**Formal analysis:** Geert van Geest.

**Methodology:** Geert van Geest.

**Resources:** Yann Haefliger.

**Software:** Yann Haefliger.

**Supervision:** Patricia M. Palagi.

**Visualization:** Geert van Geest.

**Writing – original draft:** Geert van Geest, Monique Zahn-Zabal.

**Writing – review & editing:** Geert van Geest, Monique Zahn-Zabal, Patricia M. Palagi.

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
