## [Decision Letter · Decision Letter 0]

3 Sep 2024

PONE-D-24-15760Using Glittr.org to find, compare and re-use online training materialsPLOS ONE

Dear Dr. van Geest,

Thank you for submitting your manuscript to PLOS ONE. After careful consideration, we feel that it has merit but does not fully meet PLOS ONE’s publication criteria as it currently stands. Therefore, we invite you to submit a revised version of the manuscript that addresses the points raised during the review process.

We look forward to receiving your revised manuscript.

Kind regards,

Anna Bernasconi, PhD

Academic Editor

PLOS ONE

Journal Requirements:

Additional Editor Comments:

Dear authors, thank you for submitting your research to Plos One. The work and manuscript are very interesting. Please address the minor changes suggested by reviewers 2 and 3. We will be ready and willing to reassess the revised manuscript.

Reviewers' comments:

Reviewer's Responses to Questions

**Comments to the Author**

1. Is the manuscript technically sound, and do the data support the conclusions?

Reviewer #1: Yes

Reviewer #2: Yes

Reviewer #3: Yes

2. Has the statistical analysis been performed appropriately and rigorously? 

Reviewer #1: Yes

Reviewer #2: Yes

Reviewer #3: N/A

3. Have the authors made all data underlying the findings in their manuscript fully available?

Reviewer #1: Yes

Reviewer #2: Yes

Reviewer #3: Yes

4. Is the manuscript presented in an intelligible fashion and written in standard English?

Reviewer #1: Yes

Reviewer #2: Yes

Reviewer #3: Yes

5. Review Comments to the Author

Reviewer #1: The paper is written well and the tool developed is excellent for every bioinformatician.

The UI of the tool can be more appealing.

It has 568 repositories originate 247 from 263 different GitHub/GitLab authors or organizations

Reviewer #2: Navigating the rapidly expanding field of data science and informatics can be challenging, particularly when it comes to training materials. In this context, I find that Glitter.org offers a commendable solution by serving as a metadata-lookup for public repositories and expanding adherence to the FAIR principles in open-source science. It is a valuable resource for both learning and building upon existing training materials. Additionally, the website's layout and user experience are impressive, particularly in how it segregates repositories by broader topics and provides useful filtering features based on recency (days since last commit) and user engagement (stargazers).

However, the broader success of this platform hinges on the active participation of researchers. The key challenge will be to incentivize and motivate contributors to share their repositories, turning this into a truly user-driven, widely utilized resource. While this challenge lies with the research community as a whole rather than the authors, it is an essential factor to consider.

I would recommend the publication of this work after the authors address the minor revisions and questions listed below.

1. Introduction (Page 3; Lines 46-49): What percentage of the life sciences community uses Git, and what percentage uses the listed systems? If the number is high, is there an easy way to obtain Glitter-amenable metadata from these sources as well?

2. Materials and Methods (Page 5; Lines 92-93): The term "manually curated" might give the impression that the website is static and all data has been manually added by the authors. Consider using "publicly contributed and maintained" instead.

3. Results and Discussion (Page 8; Lines 189-191): I strongly agree with this statement, and I suggest the authors highlight this point in the introduction. The resource can be useful to anyone working in informatics, data science, or machine learning.

4. Results and Discussion (Page 9; Lines 196-198): I don’t necessarily agree with this statement. In my opinion, R has the advantage of being available for longer to the informatics community. A crude search through the Glitter.org website for the "Scripting and Languages" repositories revealed the following stats for repositories with a push in the last 30 days:

- 25/93 (26.88%) in Python

- 49/247 (19.83%) in R

5. Authors and Organizations (Page 11; Lines 260-263): Consider commenting on the fact that countries incentivizing open-source science tend to take the initiative in hosting their training resources on platforms like GitHub and GitLab as well.

6. Conclusion (Page 14; Line 348): Suggest adding "in the life sciences and other informatics-driven fields" or something similar to broaden the scope.

7. Page 9 (Lines 195-198) and Page 12 (Lines 272-281): Most of the stats appear outdated in this draft. For example, there are now 389 (not 315) "Scripting and Languages" Repositories on Glitter, with 247 written in R (not 230) and 93 written in Python (not 81). I encourage the authors to double-check and update according to the latest iteration of Glitter.

Reviewer #3: In this well-written paper, the authors present Glittr.org, an aggregator of publicly available training materials in the life sciences. Glittr.org aims to enhance the FAIRness of these materials by adding and exposing metadata to them. I believe that the authors did a great job in supporting the use and re-use of Open Educational Resources and have developed a valuable tool for the life sciences education community.

Abstract:

- The abstract should clarify that users must submit the URL of their repository themselves, as leaving this information out currently implies that Glittr.org automatically scrapes repositories for resources.

- The abstract mentions metadata, but it is unclear what specific types are involved. Providing examples or elaborating on this would be preferred.

Introduction:

- The abstract mentions a focus on (bio)informatics, but the introduction references the broader life sciences. Considering the materials on Glittr.org, it might be more accurate to focus on life sciences as a whole rather than just (bio)informatics.

- The introduction mentions ‘courses’. Can only course materials be found via Glittr.org, or can other materials be submitted and accessed as well?

- ‘... aggregation and indexation, which encourages metadata standardization’: I am not sure what you are trying to state here. Are you stating that by making sure that creators of training materials are able to submit their materials to Glittr.org to be aggregated and indexed, they are encouraged to standardize their metadata, because otherwise they were not able to do so? Could you rephrase this statement?

- ‘This article introduces Glittr.org, a novel web-based resource designed to enhance the discovery and comparison of reusable bioinformatics training materials hosted on public Git repositories.’: Could you elaborate more (could be 1-2 sentences) on how the discovery and comparison is enhanced? This is described in more detail later in the manuscript, but triggers questions when reading this in the introduction.

- ‘Glittr.org contains intuitive search functionalities based on pre-defined topics and repository metadata.’: Could you state if these topics are user-inputted or defined by the Glittr.org team?

- ‘Moreover, Glittr.org supports the adherence to the FAIR principles by making all repository metadata readily accessible through REST API endpoints’: Are the REST endpoints the only adherence to the principles? I would say that the standardized metadata are also a big contributor here.

Materials and methods:

- ‘Glittr.org is a manually curated list …’: is the list fully curated or also automatically harvested? By not mentioning this in the abstract I was under the impression that Glittr.org automatically harvests resources. Moving sentences 98-100 after this sentence might help to avoid confusion.

- An example of the metadata and tags present in Glittr.org would be useful: how does the added metadata look like and how does one need to add tags?

- ‘These tags are based on existing ontologies …’: are you referring to ontology concepts here (items in an ontology)?

- ‘If a term is considered relevant, but not in any of these ontologies, a term will be used from any other ontology on BioPortal…’: Does the creator of the training material add tags manually, and if they cannot find a tag in the Glittr.org-provided list (that uses EDAM, DSEO, EFO, FDT-O) they can resort to a concept from another ontology?

- ‘Each repository is associated with one or more tags by a curator,’: Who is the curator, is this someone from the Glittr.org team?

- ‘Each tag falls within a broader category, which are maintained within Glittr.org’: How is the link between the ontology concepts and these categories maintained?

- I would expect Figure 1 to be listed under the paragraph describing the tags/metadata, not under the one describing the source code.

- Paragraphs with headings would be beneficial for readers, among which including headings for the development of the platform and how one should use it.

Results:

- It would be beneficial to include a disclaimer that repositories are manually added to Glittr.org, and the results do not reflect all available training materials in the life sciences. Now it reads more like transcriptomics is the only topic that is taught on. Statements like ‘This suggests that the R language is currently the most popular language to teach and therefore probably to learn in bioinformatics, with Python coming in second’ cannot be accurately made, as this is not a reflection of all materials in life sciences.

- Suggestion: ‘Repositories on glittr.org are categorized in six categories, as mentioned above’

- Consider using the term ‘programming languages’ instead of ‘computer languages’.

- References to the ‘Efforts that support open and reproducible science’ would be appreciated

- ‘repository with all training materials related to Galaxy’: A brief explanation of what Galaxy is would help the reader.

- ‘However, there are many communities that demand non-English bioinformatics training material, and we are open to serving these communities by listing them on Glittr.org.’: mostly out of interest, are there numbers on English materials vs. other materials?

- ‘the trainer's repository will be added (or updated) to the collection provided it meets the inclusion criteria’: Is someone from the Glittr.org team checking these criteria for every repository?

- ‘Obviously, this does not prevent the trainer from posting the material in other 14 repositories’: With 'other' it now looks like that Glittr.org is a repository, whereas earlier it's listed as an aggregator.

- Consider ‘wishes/wants’ in the sentence ‘a developer who must programmatically’

- ‘This can easily be done using the API endpoints’: Is there API documentation available, e.g. in the form of Swagger documentation?

- ‘It provides access to the metadata of all the repositories in Glittr.org in Bioschemas TrainingMaterial markup.’: this was mentioned already

- It would be useful to outline future developments or plans for Glittr.org in the manuscript.

Minor details:

The article is well-written and has a great structure. However there are a few small details that can be fixed:

- Ensure consistent usage of terms such as Git vs. git, Glittr.org vs. glittr.org, and web site vs. website.

- Fix minor grammatical issues, such as using "a URL" instead of "an URL," adding a missing closing parenthesis on line 60, and ensuring consistent use of the Oxford comma throughout the manuscript.

- Usage of ‘field’ instead of the FAIR-preferred term ‘domain’

Conclusion:

This paper presents a significant contribution to the life sciences education community through the development of Glittr.org. With minor adjustments to clarity, consistency, and additional details, the manuscript will be even stronger. The authors have created a valuable resource that enhances the FAIRness of educational materials, which is crucial for the advancement of open science and education.

6. PLOS authors have the option to publish the peer review history of their article (what does this mean?). If published, this will include your full peer review and any attached files.

Reviewer #1: **Yes: **Abhishek Appaji

Reviewer #2: **Yes: **Rishabh D. Guha

Reviewer #3: No

---

## [Author Response · Author response to Decision Letter 0]

9 Oct 2024

We thank the reviewers very much for their valuable suggestions, which significantly improved the manuscript. Because of their remarks we noticed that we had to be more precise about the scope of Glittr.org. Therefore, we slightly adjusted the title and throughout the manuscript, we replaced 'training' with the broader term 'education', and 'bioinformatics' with 'computational life sciences'. This better-defined scope will also be reflected on the Glittr.org website after the upcoming update. Below, we specifically comment on the reviewers' suggestions. 

Reviewer #1: The paper is written well and the tool developed is excellent for every bioinformatician. 

The UI of the tool can be more appealing. 

It has 568 repositories originate 247 from 263 different GitHub/GitLab authors or organizations 

We are pleased the reviewer appreciated the manuscript and the tool. The tool is intended to be simple to use, and we had hoped it would serve this purpose. We intend to organize in the future a workshop to gather information from the users on the UX and UI capabilities and adapt the tool to make it more appealing if necessary. 

Reviewer #2: Navigating the rapidly expanding field of data science and informatics can be challenging, particularly when it comes to training materials. In this context, I find that Glitter.org offers a commendable solution by serving as a metadata-lookup for public repositories and expanding adherence to the FAIR principles in open-source science. It is a valuable resource for both learning and building upon existing training materials. Additionally, the website's layout and user experience are impressive, particularly in how it segregates repositories by broader topics and provides useful filtering features based on recency (days since last commit) and user engagement (stargazers). 

However, the broader success of this platform hinges on the active participation of researchers. The key challenge will be to incentivize and motivate contributors to share their repositories, turning this into a truly user-driven, widely utilized resource. While this challenge lies with the research community as a whole rather than the authors, it is an essential factor to consider. 

We thank the reviewer for the kind words and considerations. Indeed, we also see the challenges of incentivizing users to contribute their repositories, and we see that as a major goal to enhance the sustainability of the resource. 

I would recommend the publication of this work after the authors address the minor revisions and questions listed below. 

1. Introduction (Page 3; Lines 46-49): What percentage of the life sciences community uses Git, and what percentage uses the listed systems? If the number is high, is there an easy way to obtain Glitter-amenable metadata from these sources as well? 

We thank the reviewer for these interesting questions. To our knowledge there are no available statistics on the usage by the life science community of GitHub or GitLab. However, GitHub and GitLab.com together had a market share of 70% in 2022 (https://bitrise.io/blog/post/the-state-of-mobile-app-development-in-2022) for git hosting solutions. Another big player is bitbucket, but we haven’t come across life sciences educational material on this platform. It depends very much on the platform how easy it is to use metadata for glittr.org. If there is a stable URL that points to the git repository and we can extract popularity and recency data, it would be straightforward to obtain Glittr-amenable metadata. If we find a significant number of repositories with life sciences educational materials in another git platform in the future, we aim to amend glittr to also extract metadata from there. 

2. Materials and Methods (Page 5; Lines 92-93): The term "manually curated" might give the impression that the website is static and all data has been manually added by the authors. Consider using "publicly contributed and maintained" instead. 

We thank the reviewer for the suggestion and have changed the manuscript accordingly (Page 5, Line 97-98). 

3. Results and Discussion (Page 8; Lines 189-191): I strongly agree with this statement, and I suggest the authors highlight this point in the introduction. The resource can be useful to anyone working in informatics, data science, or machine learning. 

This point has been added as well to the end of the last paragraph of the Introduction (Page 5, Line 92-94). 

4. Results and Discussion (Page 9; Lines 196-198): I don’t necessarily agree with this statement. In my opinion, R has the advantage of being available for longer to the informatics community. A crude search through the Glitter.org website for the "Scripting and Languages" repositories revealed the following stats for repositories with a push in the last 30 days: 

- 25/93 (26.88%) in Python 

- 49/247 (19.83%) in R 

The reviewer makes a good point with which we agree. The sentence has been toned down and we now only mention that R is a popular language (Page 10, Line 209-210). 

5. Authors and Organizations (Page 11; Lines 260-263): Consider commenting on the fact that countries incentivizing open-source science tend to take the initiative in hosting their training resources on platforms like GitHub and GitLab as well. 

Again, the reviewer makes a good point, which we have included in a sentence to the text (Page 12, Line 274-276). 

6. Conclusion (Page 14; Line 348): Suggest adding "in the life sciences and other informatics-driven fields" or something similar to broaden the scope. 

We agree with the reviewer. The text has been revised accordingly (Page 16, Line 359-361). 

7. Page 9 (Lines 195-198) and Page 12 (Lines 272-281): Most of the stats appear outdated in this draft. For example, there are now 389 (not 315) "Scripting and Languages" Repositories on Glitter, with 247 written in R (not 230) and 93 written in Python (not 81). I encourage the authors to double-check and update according to the latest iteration of Glitter. 

Indeed, things move fast in the field. We have updated this information just before resubmission. 

Reviewer #3: In this well-written paper, the authors present Glittr.org, an aggregator of publicly available training materials in the life sciences. Glittr.org aims to enhance the FAIRness of these materials by adding and exposing metadata to them. I believe that the authors did a great job in supporting the use and re-use of Open Educational Resources and have developed a valuable tool for the life sciences education community. 

We thank the reviewer for such nice feedback. We are pleased the reviewer appreciated the tool and our efforts to promote open and FAIR training materials. 

Abstract: 

- The abstract should clarify that users must submit the URL of their repository themselves, as leaving this information out currently implies that Glittr.org automatically scrapes repositories for resources. 

Thank you for this suggestion. The abstract was restructured, and we added the sentence” Glittr.org is a manually curated database of Git repositories” to emphasize this point (Page 2, Line 16-17). 

- The abstract mentions metadata, but it is unclear what specific types are involved. Providing examples or elaborating on this would be preferred. 

We agree with the reviewer. This has been moved to the introduction (Page 5, Line 90) and further elaborated in the Material and Methods (Page 5, Line 106-108). 

Introduction: 

- The abstract mentions a focus on (bio)informatics, but the introduction references the broader life sciences. Considering the materials on Glittr.org, it might be more accurate to focus on life sciences as a whole rather than just (bio)informatics. 

We fully agree with the reviewer and have changed the scope to computational life sciences (e.g. in line 14, page 2; line 85 -86, page 4 etc.). 

- The introduction mentions ‘courses’. Can only course materials be found via Glittr.org, or can other materials be submitted and accessed as well? 

The focus of glittr.org is educational material, so it covers courses, modules or lessons. However the framework can be used for any collection of git repositories. 

- ‘... aggregation and indexation, which encourages metadata standardization’: I am not sure what you are trying to state here. Are you stating that by making sure that creators of training materials are able to submit their materials to Glittr.org to be aggregated and indexed, they are encouraged to standardize their metadata, because otherwise they were not able to do so? Could you rephrase this statement? 

We agree with the reviewer. The sentence has been reworded and is hopefully clearer (Page 3, Line 46-49). 

- ‘This article introduces Glittr.org, a novel web-based resource designed to enhance the discovery and comparison of reusable bioinformatics training materials hosted on public Git repositories.’: Could you elaborate more (could be 1-2 sentences) on how the discovery and comparison is enhanced? This is described in more detail later in the manuscript, but triggers questions when reading this in the introduction. 

We agree with the reviewer. We have modified the paragraph to make this (hopefully) clearer (Page 4, Line 84-94).. 

- ‘Glittr.org contains intuitive search functionalities based on pre-defined topics and repository metadata.’: Could you state if these topics are user-inputted or defined by the Glittr.org team? 

We agree with the reviewer. This is now specified in the sentence ‘This database of Git repositories is publicly maintained‘ (Page 5, Line 86-89). 

- ‘Moreover, Glittr.org supports the adherence to the FAIR principles by making all repository metadata readily accessible through REST API endpoints’: Are the REST endpoints the only adherence to the principles? I would say that the standardized metadata are also a big contributor here. 

Quite true. We have revised the paragraph to make this clearer (Page 5, Line 89-92). 

Materials and methods: 

- ‘Glittr.org is a manually curated list …’: is the list fully curated or also automatically harvested? By not mentioning this in the abstract I was under the impression that Glittr.org automatically harvests resources. Moving sentences 98-100 after this sentence might help to avoid confusion. 

We agree with the reviewer. This has been added to the abstract (Page 2, Line 16-17). 

- An example of the metadata and tags present in Glittr.org would be useful: how does the added metadata look like and how does one need to add tags? 

We agree with the reviewer. We’ve extended the material and methods to address this comment (Page 6, Line 124-126). 

- ‘These tags are based on existing ontologies …’: are you referring to ontology concepts here (items in an ontology)? 

Yes, we were referring to items (terms) in an ontology. The text has been revised (Page 6, Line 112-114). 

- ‘If a term is considered relevant, but not in any of these ontologies, a term will be used from any other ontology on BioPortal…’: Does the creator of the training material add tags manually, and if they cannot find a tag in the Glittr.org-provided list (that uses EDAM, DSEO, EFO, FDT-O) they can resort to a concept from another ontology? 

Yes, we try to be as flexible as possible regarding these terms. Therefore, a contributor can request terms/tags that are not yet available in glittr.org. The text has been revised to state this explicitly (Page 6, Line 124-126). 

- ‘Each repository is associated with one or more tags by a curator,’: Who is the curator, is this someone from the Glittr.org team? 

Yes, the curator is part of the Glittr.org team. Again, the text has been revised to state this explicitly (Page 6, Line 128-129). 

- ‘Each tag falls within a broader category, which are maintained within Glittr.org’: How is the link between the ontology concepts and these categories maintained? 

This is done by the Glittr.org team. The text has been revised to state this explicitly (Page 6, Line 135-136). 

- I would expect Figure 1 to be listed under the paragraph describing the tags/metadata, not under the one describing the source code. 

We agree with the reviewer. The position of Figure 1 has been modified. The text has been revised to state this explicitly (Page 7, Line 141-143). 

- Paragraphs with headings would be beneficial for readers, among which including headings for the development of the platform and how one should use it. 

We agree with the reviewer. We have divided the materials and method section into more subchapter to improve readability. 

Results: 

- It would be beneficial to include a disclaimer that repositories are manually added to Glittr.org, and the results do not reflect all available training materials in the life sciences. Now it reads more like transcriptomics is the only topic that is taught on. Statements like ‘This suggests that the R language is currently the most popular language to teach and therefore probably to learn in bioinformatics, with Python coming in second’ cannot be accurately made, as this is not a reflection of all materials in life sciences. 

Agreed, the statement was too emphatic. We’ve modified the text. We’ve also added a sentence to the introductory paragraph of ‘Training material topics’ what this overview is about (Page 10, Line 207-227) 

- Suggestion: ‘Repositories on glittr.org are categorized in six categories, as mentioned above’ 

 We agree with the reviewer. We adapted the text accordingly (Page 9, Line 200. 

- Consider using the term ‘programming languages’ instead of ‘computer languages’. 

 We agree with the reviewer. We adapted the text accordingly (Page 9, Line 202). 

- References to the ‘Efforts that support open and reproducible science’ would be appreciated 

We agree with the reviewer. References to the RDM community and EOSC have been included (Page 11, Line 233). 

- ‘repository with all training materials related to Galaxy’: A brief explanation of what Galaxy is would help the reader. 

We agree with the reviewer. A brief explanation has been added to the text (Page 11, Line 254). 

- ‘However, there are many communities that demand non-English bioinformatics training material, and we are open to serving these communities by listing them on Glittr.org.’: mostly out of interest, are there numbers on English materials vs. other materials? 

We currently do not track these. But there are no more than 10 non-English repos in glittr.org currently, with most of them in either French or Spanish. 

- ‘the trainer's repository will be added (or updated) to the collection provided it meets the inclusion criteria’: Is someone from the Glittr.org team checking these criteria for every repository? 

Yes, we’ve adjusted the text to make this clearer (Page 15, Line 332-334). 

- ‘Obviously, this does not prevent the trainer from posting the material in other 14 repositories’: With 'other' it now looks like that Glittr.org is a repository, whereas earlier it's listed as an aggregator. 

We agree with the reviewer, and replaced ‘repositories’ with registries (Page 15, Line 336). 

- Consider ‘wishes/wants’ in the sentence ‘a developer who must programmatically’ 

We agree with the reviewer, and revised the text (Page 15, Line 340). 

- ‘This can easily be done using the API endpoints’: Is there API documentation available, e.g. in the form of Swagger documentation? 

The api documentation is in the GitHub readme. Although considered valuable, there is no Swagger documentation implemented at the moment. 

- ‘It provides access to the metadata of all the repositories in Glittr.org in Bioschemas TrainingMaterial markup.’: this was mentioned already 

We agree with the reviewer. This part has been shortened (Page 15, Line 344-346). 

- It would be useful to outline future developments or plans for Glittr.org in the manuscript. 

Thank you for this suggestion. We have included a sentence in the conclusion (page 16, line 357 – 358) 

Minor details: 

The article is well-written and has a great structure. However there are a few small details that can be fixed: 

- Ensure consistent usage

---

## [Decision Letter · Decision Letter 1]

13 Nov 2024

Using Glittr.org to find, compare and re-use onlinematerials for training and education

PONE-D-24-15760R1

Dear Dr. van Geest,

We’re pleased to inform you that your manuscript has been judged scientifically suitable for publication and will be formally accepted for publication once it meets all outstanding technical requirements.

Kind regards,

Anna Bernasconi, PhD

Academic Editor

PLOS ONE

Additional Editor Comments (optional):

Dear authors,

we are happy to recommend publication of your article after the first round of revision, having addressed all concerns from the reviewers. Please, when preparing the proof version, mind the comment from Rev#3: "One final detail to address before the manuscript is ready for publication is updating the repository count. While the results section lists 664 repositories, the conclusion still references the previous count (500+)."

Reviewers' comments:

Reviewer's Responses to Questions

**Comments to the Author**

1. If the authors have adequately addressed your comments raised in a previous round of review and you feel that this manuscript is now acceptable for publication, you may indicate that here to bypass the “Comments to the Author” section, enter your conflict of interest statement in the “Confidential to Editor” section, and submit your "Accept" recommendation.

Reviewer #2: All comments have been addressed

Reviewer #3: All comments have been addressed

2. Is the manuscript technically sound, and do the data support the conclusions?

Reviewer #2: Yes

Reviewer #3: Yes

3. Has the statistical analysis been performed appropriately and rigorously? 

Reviewer #2: Yes

Reviewer #3: N/A

4. Have the authors made all data underlying the findings in their manuscript fully available?

Reviewer #2: Yes

Reviewer #3: (No Response)

5. Is the manuscript presented in an intelligible fashion and written in standard English?

Reviewer #2: Yes

Reviewer #3: (No Response)

6. Review Comments to the Author

Reviewer #2: The authors have done a commendable job of addressing the comments and suggestions of all the three reviewers. I would recommend the publication of this article in its revised form.

Reviewer #3: Thank you for addressing the comments of all reviewers.

One final detail to address before the manuscript is ready for publication is updating the repository count. While the results section lists 664 repositories, the conclusion still references the previous count (500+).

7. PLOS authors have the option to publish the peer review history of their article (what does this mean?). If published, this will include your full peer review and any attached files.

Reviewer #2: **Yes: **Rishabh D. Guha

Reviewer #3: No

---

## [Editor Report · Acceptance letter]

25 Nov 2024

PONE-D-24-15760R1 

PLOS ONE

Dear Dr. van Geest, 

I'm pleased to inform you that your manuscript has been deemed suitable for publication in PLOS ONE. Congratulations! Your manuscript is now being handed over to our production team.

Kind regards, 

on behalf of

Dr. Anna Bernasconi 

Academic Editor

PLOS ONE